# Navigating the Alzheimer’s Biomarker Landscape: A Comprehensive Analysis of Fluid-Based Diagnostics

**DOI:** 10.3390/cells13221901

**Published:** 2024-11-18

**Authors:** Elsa El Abiad, Ali Al-Kuwari, Ubaida Al-Aani, Yaqoub Al Jaidah, Ali Chaari

**Affiliations:** Weill Cornell Medicine–Qatar, Qatar Foundation, Education City, Doha P.O. Box 24144, Qatar; ela4008@qatar-med.cornell.edu (E.E.A.); aka4006@qatar-med.cornell.edu (A.A.-K.); uba4001@qatar-med.cornell.edu (U.A.-A.); yja4001@qatar-med.cornell.edu (Y.A.J.)

**Keywords:** Alzheimer’s disease (AD), cerebrospinal fluid (CSF), saliva, blood, artificial intelligence (AI)

## Abstract

Background: Alzheimer’s disease (AD) affects a significant portion of the aging population, presenting a serious challenge due to the limited availability of effective therapies during its progression. The disease advances rapidly, underscoring the need for early diagnosis and the application of preventative measures. Current diagnostic methods for AD are often expensive and invasive, restricting access for the general public. One potential solution is the use of biomarkers, which can facilitate early detection and treatment through objective, non-invasive, and cost-effective evaluations of AD. This review critically investigates the function and role of biofluid biomarkers in detecting AD, with a specific focus on cerebrospinal fluid (CSF), blood-based, and saliva biomarkers. Results: CSF biomarkers have demonstrated potential for accurate diagnosis and valuable prognostic insights, while blood biomarkers offer a minimally invasive and cost-effective approach for diagnosing cognitive issues. However, while current biomarkers for AD show significant potential, none have yet achieved the precision needed to replace expensive PET scans and CSF assays. The lack of a single accurate biomarker underscores the need for further research to identify novel or combined biomarkers to enhance the clinical efficacy of existing diagnostic tests. In this context, artificial intelligence (AI) and deep-learning (DL) tools present promising avenues for improving biomarker analysis and interpretation, enabling more precise and timely diagnoses. Conclusions: Further research is essential to confirm the utility of all AD biomarkers in clinical settings. Combining biomarker data with AI tools offers a promising path toward revolutionizing the personalized characterization and early diagnosis of AD symptoms.

## 1. Introduction

Neurodegenerative diseases (NDDs) represent a diverse group of neurological disorders that progressively result in the loss of neurons in the central or peripheral nervous systems, significantly affecting the lives of millions globally [1]. As life expectancy continues to rise, the prevalence of NDDs is projected to increase, with dementia cases expected to reach 150 million by 2050.

Alzheimer’s disease (AD) is the most common form of NDD and has a significant impact on individuals, families, and healthcare systems [2]. Healthcare costs associated with AD are anticipated to exceed USD 1 trillion as the population ages, driven by the growing number of elderly vulnerable to the disease [3]. This escalating burden highlights the urgent need to understand the progression of AD.

Delays in seeking medical assistance can hinder timely intervention, making early detection crucial [3]. Research emphasizes the importance of continuous monitoring for effective dementia treatment [3], as it typically takes an average of 5.5 years from the onset of symptoms to receive an AD diagnosis [4].

AD unfolds in distinct stages, each characterized by specific physiological markers: presymptomatic, Mild Cognitive Impairment (MCI), and moderate to severe dementia [5]. A key indicator of disease progression is Amyloid Beta (Aβ) aggregation, which results from incorrect cleavage of Amyloid Precursor Protein (APP) [6]. This process produces hydrophobic peptides that form plaques, causing neuronal damage and inflammatory responses [6,7]. As AD advances, cognitive decline becomes more pronounced, with disorientation and memory difficulties setting in, brought on by neurofibrillary tangles and Amyloid Beta (Aβ) plaques [8]. In the final stages, patients experience severe symptoms, including behavioral changes and the loss of personal memory [5].

The rising incidence and economic burden of AD highlight the urgent need for a comprehensive strategy to address this illness. One major challenge is the high cost associated with diagnostic imaging techniques such as Positron Emission Tomography (PET) and magnetic resonance imaging (MRI), which can prevent individuals from seeking early AD detection [5,9,10]. PET scans, including technical tests and professional fees, often cost between USD and USD 1400 [11]. Additionally, the sensitivity and specificity of traditional clinical techniques are limited, making the accurate diagnosis of AD challenging. For instance, cortical thinning observed in MRI scans may be caused by other medical conditions, while amyloid plaques, detected through PET scans, can also appear in individuals without AD, leading to inaccurate diagnoses [12,13]. Furthermore, current imaging methods often miss opportunities for early intervention, as they typically diagnose the disease only after symptoms have already emerged. Since early detection has the potential to slow the progression of AD [14], it is justified to explore alternative diagnostic techniques that offer greater accuracy, specificity, efficiency, and accessibility, which is crucial.

To address this gap, biomarkers have emerged as a pivotal advancement in AD diagnosis and management (Figure 1). These molecular indicators, detectable in cerebrospinal fluid, blood, and saliva, offer valuable insights into AD pathology, potentially before the onset of symptoms [15,16]. By enabling early detection and precise monitoring of disease progression, biomarkers have the potential to transform clinical practice, paving the way for more timely interventions and personalized treatment strategies for AD patients [15,16]. Their integration into routine medical care represents a paradigm shift in our approach to combating this neurodegenerative disorder.

This review provides an analysis of fluid-based biomarkers, including CSF, blood, and saliva. We highlight the specific markers and proteins.

## 2. Potential Biomarkers for AD

### 2.1. Cerebrospinal Fluid Biomarkers

#### 2.1.1. Background

CSF is a clear, colorless liquid produced by the brain’s ventricles that circulates throughout the brain and spinal cord [17,18]. Its primary function is to protect the central nervous system by providing cushioning while also supplying nutrients and removing waste products from the brain [17,19]. Historically, CSF has been used as a diagnostic marker for infections and neurological disorders since the 16th century [17]. Today, CSF biomarkers hold great potential in the diagnosis and understanding of various NDDs, including AD. By analyzing the specific biomarkers present in CSF, researchers and clinicians can gain valuable insights into the disease’s progression and underlying mechanisms, offering promising avenues for early detection and improved treatment strategies.

#### 2.1.2. Findings

A number of markers have been identified within CSF. These range from Amyloid proteins and tau to various axonal damage markers. Each marker indicates specific pathologies or changes associated with AD onset.

#### 2.1.3. Biomarkers


A. BETA-AMYLOID (Aβ42):


Aβ comes in two primary isoforms associated with AD: Aβ40 and Aβ42, which differ in their number of amino acid residues [20]. Aβ42 is more prone to aggregation than Aβ40, and it is this tendency to form insoluble amyloid plaques in the brain that plays a central role in AD progression. As a result of the aggregation of Aβ42 into plaques, its levels in CSF are significantly reduced in AD individuals compared to non-AD individuals. This reduction makes Aβ42 a particularly valuable biomarker for early diagnosis and monitoring of AD [21]. Typically, Aβ42 concentrations in AD patients are below 1000 pg/mL, while non-AD individuals have levels above 1700 pg/mL [22,23]. The reduction in Aβ42 in CSF allows for the differentiation of AD individuals from healthy individuals, with a sensitivity of 90% and a specificity of 80% [24]. This reduction also enables the distinction between frontotemporal dementia and AD, with a sensitivity of 88.8% and a specificity of 80% [25,26]. Overall, these studies demonstrate that Aβ42 in CSF is a strong candidate as a reliable biomarker for AD.


B. Total tau (t-tau):


In AD patients, elevated t-tau levels and aggregation are directly linked to neural damage and cell death, making t-tau an important biomarker for assessing disease severity and progression [27,28]. Tau is a microtubule-associated protein predominantly found in the axons of neurons, where it plays a critical role in microtubule stabilization [29]. Research suggests that elevated t-tau protein levels in CSF reflect the extent of neuronal damage, as they are closely associated with neurofibrillary tangles, a hallmark of NDDs, including AD [30]. In healthy individuals aged 21 to 50 years, t-tau levels are typically below 300 pg/mL, gradually increasing to around 500 pg/mL as they age. In contrast, t-tau levels in AD patients typically range from 300 to 900 pg/mL, depending on age and disease progression [31,32,33]. Given these elevated levels, t-tau has proven to be a reliable biomarker for distinguishing AD patients from healthy individuals, achieving a sensitivity of 84% and a specificity of 91% [34]. This highlights the potential of t-tau as a valuable biomarker for both diagnosis and monitoring of AD progression.


C. Phosphorylated tau (p-tau):


Phosphorylated tau (p-tau) levels are higher in AD patients compared to healthy individuals [35]. Tau hyperphosphorylation can lead to pathological conditions such as neuronal degeneration and the formation of neurofibrillary tangles, both hallmarks of AD [30,35]. In healthy individuals, p-tau levels typically measure 27.09 ± 7.18 pg/mL, whereas in AD patients, levels are around 67.87 ± 18.05 pg/mL [35,36,37]. The sensitivity and specificity of tau biomarkers are crucial for their effectiveness in diagnosing and monitoring AD. Studies have demonstrated that p-tau shows high sensitivity (90.2%) and specificity (80%) in differentiating AD from other non-AD diseases [38]. Additionally, when specifically analyzing the p-tau to t-tau ratio, sensitivity increases to 95%, and specificity ranges between 86 and 100% depending on the group tested, providing even greater diagnostic accuracy [39].


D. Irisin:


Irisin is a hormone that enhances learning and memory by promoting brain-derived neurotrophic factor (BDNF), suppressing neuroinflammation, and improving insulin resistance and glucose homeostasis [40]. Research indicates that irisin levels in CSF are reduced in AD patients compared to healthy individuals, suggesting its potential as both a biomarker and therapeutic target for AD [41,42]. Notably, CSF irisin shows a positive correlation with Aβ42, a well-established AD biomarker [41]. There is also evidence of a negative trend between CSF irisin, t-tau levels, and other AD biomarkers, although further studies are needed to confirm this relationship [41,42]. Overall, irisin shows promise as a biomarker for AD, but more research is needed to understand its role in diagnosis.


E. Neurofilament Light Chain (NfL):


Elevated levels of neurofilament light (NfL) in CSF are associated with the severity and progression of AD and can help distinguish AD from other types of dementia, making NfL a strong candidate for assessing the extent of neurodegeneration in AD patients. NfL is an essential structural component of the neuronal cytoskeleton, and it is released into the CSF and blood when axons are damaged or undergo degeneration [43,44,45]. In AD patients, NfL levels are typically around 45.9 pg/mL, compared to 32.1 pg/mL in healthy controls [46]. However, for diagnosing AD, the sensitivity of NfL is 59.6%, and the specificity is 76.2% [47], indicating that further research is needed to fully assess its potential as a reliable biomarker.


F. Synaptic and Axonal Damage Markers:


Neurogranin, a postsynaptic protein, is another biomarker candidate for AD, primarily due to its role in reflecting synaptic degeneration and cognitive decline [46,48,49]. Elevated levels of neurogranin have been detected in the CSF of AD patients, with concentrations ranging from 336 to 382 pg/mL [46,48,49]. Research has found varying degrees of correlation between neurogranin levels and other key AD biomarkers, including t-tau, p-tau, and the Aβ42/40 ratio, which are crucial indicators of the disease’s progression [50]. Additionally, neurogranin has demonstrated a sensitivity and specificity of 0.73 for distinguishing AD from other neurodegenerative diseases and from healthy control groups [50].


G. Inflammatory Markers:


Inflammation plays a critical role in the pathophysiology of AD [51], positioning inflammatory markers as potential candidates for AD diagnosis. Two notable candidates are chitinase-3-like protein 1 (YKL-40) and interleukin-6 (IL-6).

YKL-40 is a glycoprotein produced by astrocytes, and its elevated production may indicate neuroinflammation [52]. In AD patients, YKL-40 levels are elevated in CSF and correlate positively with the severity and progression of the disease [52]. While no specific threshold for YKL-40 levels indicative of AD has been established, studies show that it often correlates with t-tau and p-tau [53,54]. In one study, CSF YKL-40 demonstrated a sensitivity of 65.6% and a specificity of 66.3% in distinguishing AD from healthy individuals and other NDDs [55].

Elevated levels of the pro-inflammatory cytokine IL-6 in CSF have also been linked to AD [56], with one study reporting a sensitivity of 76.9% and a specificity of 100% in predicting AD [16]. While IL-6 levels above 10 pg/mL are typically associated with inflammatory neurological disorders, their specific relevance to AD requires further validation [57].

#### 2.1.4. Limitations of CSF Biomarkers in Diagnosing AD

CSF is collected from the spinal cord via a lumbar puncture, a procedure that, while generally safe, carries risks due to its invasiveness. Potential complications include more serious effects, such as seizures, bleeding, infection, and headaches [58,59]. A total of 53.3% of participants in one study reported experiencing headaches after CSF catheterization, with 33.3% requiring blood patches [60]. In addition to safety concerns, logistical challenges also hinder the widespread medical use of CSF testing. Standardized procedures and assays are required to ensure accurate and consistent readings across different laboratories [59].

Although CSF testing can help distinguish AD from other types of dementia, many biomarkers—aside from Aβ40/42—are not unique to AD. In this context, the CSF patterns of different NDDs may sometimes be indistinguishable, complicating the diagnostic process [59]. Table 1 provides an overview of the aforementioned CSF-derived markers.

### 2.2. Blood-Based Biomarkers

#### 2.2.1. Background

Currently, patients often undergo costly diagnostic procedures, such as amyloid PET scans or invasive CSF testing, to diagnose AD. In response, extensive research has focused on developing blood-based biomarkers as a less invasive alternative [62]. Blood samples contain numerous biomarkers that could be linked to AD pathology. Successfully identifying these biomarkers could enable earlier detection of AD, minimizing the need for invasive diagnostic procedures. Early diagnosis is essential, as it allows for timely intervention, which is critical for improving long-term outcomes in patients [63]. However, despite these advancements, current blood-based biomarkers are not yet reliable enough to be used as standalone diagnostic tools for AD [64]. Further developments are necessary before they can replace more invasive methods. Below, we outline the main blood-based biomarkers discovered to date.

#### 2.2.2. Findings


A. AD Linked Marker Genes:


AD is driven by a complex array of factors, including genetics, family history, and advanced age. Family and twin studies suggest that genetic factors contribute to approximately 80% of AD cases [65]. Several genes are involved in increasing the risk of early-onset AD, most notably the APP, PSEN1, and PSEN2 genes [66]. Polymorphisms in the APOE gene also contribute to a higher incidence of AD [67].

APP is a transmembrane protein that is highly expressed in neurons within the central and peripheral nervous systems [68]. APP functions as a receptor and plays a role in synapse formation and neuronal plasticity [69]. Although several mutations in the genes coding for APP have been identified, the mechanisms by which these mutations contribute to Aβ pathology remain unclear. However, research has shown that specific mutations, such as A673V, E682K, and E693Q, increase the Aβ42/Aβ40 ratio, which is closely linked to AD progression [70].

From a clinical perspective, studies have detected recombinant forms of APP in blood plasma, suggesting that it could potentially serve as a biomarker of AD. From a clinical perspective, research has identified recombinant forms of APP in blood plasma, indicating that APP could potentially serve as a non-invasive biomarker for AD. This discovery is particularly important because APP fragments in the bloodstream are linked to neuronal death; when brain cells are damaged, they release APP fragments into the plasma, allowing them to be detected in blood samples [71].

Another study investigated the potential of using APP in platelets as a marker for AD. The research specifically focused on two APP protein isoforms (130 KDa and 106–110 KDa), which were significantly lower in patients with mild cognitive impairment (MCI) compared to healthy individuals. Since MCI is often considered a precursor to AD, changes in these isoform ratios could serve as an early warning sign for the progression from MCI to AD, offering a valuable diagnostic tool [72].

The PSEN1 gene, located on chromosome 14, encodes presenilin-1, a key component of the *γ*-secretase complex that processes APP [73]. Several pathogenic mutations in this gene have been identified in AD. These mutations alter the ratios of Aβ42/Aβ40 by causing variations in APP cleavage [74]. Some mutations also result in a loss of enzyme function, promoting tau hyperphosphorylation [75], another hallmark of AD. Additionally, certain mutations in PSEN1 contribute to AD pathology through apoptotic mechanisms [76]. From a clinical perspective, research on PSEN1 mutations has shown that individuals carrying variants affecting the transmembrane domain of the presenilin-1 protein experience greater cognitive decline, smaller hippocampal volume, and higher levels of phosphorylated tau, underscoring the relevance of these mutations in AD progression [77].

The PSEN2 gene, located on chromosome 1, encodes presenilin-2, a transmembrane protein that shares 67% similarity with the presenilin-1 protein [78]. Presenilin-2 is also part of the *γ*-secretase complex, which plays a critical role in processing APP into amyloid-β peptides, a process central to Alzheimer’s disease (AD) pathology. However, compared to PSEN1, fewer AD-linked mutations have been identified in PSEN2. Further analysis has shown that only around 50% of these mutations meet the established criteria for pathogenicity, while the remaining mutations are classified as variants of uncertain significance (VUS) [79]. Understanding the specific role of PSEN2 mutations in AD progression remains an ongoing area of research, with important implications for both diagnosis and potential treatment development.

The ApoE gene encodes a glycoprotein primarily responsible for transporting cholesterol and phospholipids. This protein is abundant in the central nervous system and plays a significant role in the pathogenesis of AD [80]. The most relevant ApoE isoform is ApoƐ4, which is highly expressed in cells undergoing stress or aging [81]. ApoƐ4 impairs Aβ clearance, promotes lipid accumulation in microglia by inhibiting the ApoE-TREM2-PLC*γ*2 pathway, and induces tau fibril formation [82]. Individuals who carry this isoform are at significantly higher risk of developing late-onset AD (LOAD), with heterozygous carriers having a 3–4 times higher risk and homozygous carriers having a 9–15 times higher risk [83]. A large genetic association study involving over 68,000 individuals further confirmed the strong correlation between the presence of ApoƐ4 and the incidence of LOAD [84]. Additionally, ApoƐ4 has been linked to other AD markers, with carriers showing lower Aβ42/Aβ40 ratios and elevated levels of p-tau and Glial Fibrillary Acidic Protein (GFAP) [85]. These associations further underscore ApoƐ4’s role in the disease process and its potential utility as a predictive marker for AD progression.


B. Aβ42/Aβ40:


One of the key pathological markers for AD is the accumulation of Aβ protein, which leads to a reduction in circulating Aβ levels. In addition to CSF, plasma Aβ42/Aβ40 ratio reductions can also be used as blood biomarkers to indicate amyloid pathology in AD [86,87]. A study demonstrated statistically significant reductions in plasma Aβ42/Aβ40 ratios at the group level between individuals with positive amyloid PET scan results and those without [88].

Plasma Aβ42/Aβ40 ratios have demonstrated high accuracy in distinguishing AD patients from healthy controls, achieving a sensitivity of 96% and a specificity of 83% [89].

Given the central role of amyloid in AD pathology, alongside other key proteins like tau, ongoing advances in Aβ biomarker detection are critical for improving early diagnosis and treatment strategies.


C. P-tau:


In the early stages of AD, plasma p-tau levels progressively rise over time. In the brains of AD patients, P-tau plays a critical role in the formation of neurofibrillary tangles [90]. Among the various forms of p-tau, p-tau217 is notably elevated during both the preclinical and prodromal phases of AD, making it a valuable biomarker for early detection [88]. Highly sensitive assays have been developed to measure p-tau biomarkers, particularly focusing on p-tau181, p-tau217, and p-tau231 [37]. Research revealed that p-tau181 can predict AD up to eight years before pathogenic structural changes are confirmed in brain tissue post-mortem [63]. Clinically, p-tau 217 has been shown to have higher diagnostic accuracy than other plasma biomarkers, and its levels are comparable to CSF [64,91,92,93].

Additionally, changes in plasma p-tau181, p-tau217, and p-tau231 align with the onset of abnormalities in amyloid-PET imaging, with p-tau231 often presenting alterations earlier than the other p-tau biomarkers [88].


D. Plasma NfL:


NfL are key components of the cellular cytoskeleton in neuro-axonal compartments. NfL levels rise in plasma and CSF in response to neuronal damage [94]. Studies on individuals with AD mutations, such as PSEN1, have shown that elevated NfL levels can signal neurodegeneration up to ten years before clinical symptoms appear [95]. Elevated NfL levels in plasma have been linked to both subjective cognitive decline and MCI, correlating with poor memory performance and reflecting AD pathology similarly to CSF levels [44,96].

Clinical application of NfL as a biomarker for AD diagnosis has yielded promising results, with a sensitivity of 84% and a specificity of 78% [97]. However, while elevated NfL levels are indicative of neuronal damage, they are not exclusive to AD, as increased levels have also been observed in other NDDs [16].


E. Plasma Glial Fibrillary Acidic Protein:


GFAP is a protein involved in maintaining cell structure, regulating cell movement, and supporting the BBB [98]. Predominantly expressed in brain astrocytes, GFAP is a well-established marker of astrocyte activation [99]. AD individuals exhibit higher concentrations of GFAP in both CSF and blood compared to cognitively unimpaired (CU) individuals [63].

Plasma GFAP has shown greater accuracy than CSF GFAP in distinguishing between Aβ+ and Aβ− individuals and has also been linked to the presence of tau in those with AD [100]. Additional research has revealed that GFAP levels rise before other biomarkers (i.e., p-tau and NfL), suggesting that GFAP may be an early indicator of AD pathology, even before tangle formation and widespread neurodegeneration occur [101].

A small-scale study found that blood GFAP could distinguish AD from frontotemporal lobar degeneration (FTLD) patients, achieving a sensitivity of 89% and a specificity of 79% [63]. In individuals with MCI, plasma GFAP levels can predict the development and cognitive decline of AD, as well as cognitive decline in CU individuals [102]. In addition to GFAP, other structures that act as biomarkers of AD are also linked to inflammatory mechanisms.


F. sTREM2:


Soluble Triggering Receptor Expressed on Myeloid Cells 2 (sTREM2) plays a crucial role in regulating microglial clearance of Aβ, inflammatory signaling, and cell survival [103]. Elevated levels of sTREM2 indicate an inflammatory response commonly seen in AD and microglial infiltration [104]. Studies have shown that plasma sTREM2 levels peak in MCI patients, with further research linking these elevated levels to the conversion from MCI to AD and faster cognitive decline [105,106]. Additionally, sTREM2 has been significantly correlated with cerebrospinal fluid (CSF) Aβ42 in AD patients, though it does not show a strong association with t-tau or p-tau [63]. Other studies have identified positive relationships between plasma sTREM2 levels and white matter hyperintensities, as well as CSF NfL levels, suggesting a broader connection to neuroinflammation and neurodegeneration in AD [107].


G. YKL-40:


YKL-40 is a glycoprotein that has emerged as a potential biomarker. Elevated levels of YKL-40 have been associated with several adverse outcomes, including reduced brain volume, poorer cognitive performance, and an increased risk of developing dementia [108]. These findings suggest that YKL-40 may play a role in neuroinflammation and the progression of AD. However, despite its promise as a biomarker, YKL-40 is still considered novel, and more extensive research is required to fully understand its role in detecting and monitoring disease progression.

#### 2.2.3. Limitations of Blood-Based Biomarkers in Diagnosing AD

Recent research indicates that blood-based biomarker levels for AD can be influenced by demographic factors such as age, race, and ethnicity. Addressing these requires evaluating a diverse participant sample to ensure more accurate assessments [64].

Using plasma Aβ42/Aβ40 ratios to detect amyloid-β (Aβ) pathology presents challenges. While cerebrospinal fluid (CSF) shows a 40–60% drop in Aβ42/Aβ40 ratios in Aβ-positive individuals, the reduction in plasma is only 8–15%, limiting its sensitivity as a biomarker [88].

The Global Biomarker Standardization Consortium discovered a poor correlation between 11 different plasma Aβ42/40 assays; the results demonstrated that mass spectrometry-based techniques were more accurate than the majority of immunoassays in identifying brain Aβ pathology [64].

Studies investigating various lifestyle variables affecting blood biomarkers have shown that specific medications can have a significant impact on biomarker values [109]. The microglial-modulating drug minocycline has been shown to increase plasma NfL concentration severalfold [110], establishing the possibility of conflating these increases with those linked to AD pathogenesis. A separate drug, the anti-heart failure medication neprilysin, decreased plasma Aβ42/Aβ40 levels by nearly 30% [111]. Nutritional plans also play a role, wherein standardized meals have been shown to reduce NfL, GFAP, and P-tau 181 and 231 by up to 20% within hours [112], unintentionally impacting lab results and leading to the possibility of misdiagnosing or mis-staging of the disease. Table 2 provides an overview of the aforementioned blood-derived markers.

#### 2.2.4. Clinical Implementation

Large-scale cohort studies have demonstrated the growing potential of blood-based biomarkers for the diagnosis and monitoring of AD. For instance, a comprehensive study involving 2277 individuals demonstrated the applicability of several plasma biomarkers, such as the Aβ42/40 ratio, NfL, and t-tau protein [119]. The study found strong correlations between these blood-based markers and CSF biomarkers. Notably, elevated levels of p181-tau and NfL in both plasma and CSF have been found linked to faster progression from MCI to AD, indicating their importance in tracking disease advancement [119].

Research has shown that plasma panels focusing on specific amyloid-beta (Aβ) isoforms and the genetic marker Apolipoprotein E ε4 (ApoƐ4) are among the most predictive tools for determining amyloid plaques in PET scans [97]. This breakthrough is part of a larger effort in the development of biomarkers for AD diagnostics, following a 5-phase framework that was adapted from oncology diagnostics [120] (Figure 2). Phase 1 focuses on preclinical exploratory testing aiming to identify novel biomarkers for AD detection. Phase 2 moves on to developing clinical assays, which assess the rate of true and false positives associated with these biomarkers. Phase 3 is the performance of longitudinal studies designed to evaluate the predictive power of the biomarkers at various stages of AD progression. In phase 4, prospective diagnostic accuracy studies are conducted to measure the sensitivity and specificity of the biomarkers in clinical practice. Phase 5 examines the morbidity and mortality reductions, assessing how applying these biomarkers in clinical settings can reduce the overall disease burden for patients [121].

While many biomarkers do not yet meet the standards required for stand-alone screening tests, phase 4 and phase 5 studies have the potential to offer data for their deployment in triage assessments [122]. Despite the significant advancements made in recent years, further efforts are necessary to achieve widespread clinical implementation of these markers.

### 2.3. Saliva Biomarkers

#### 2.3.1. Background

Saliva plays a crucial role in protecting against the invasion of various pathogens (reference). Compared to other biofluids, saliva is a desirable option for AD detection due to several advantages: it is easy to collect, convenient, non-invasive, does not require anticoagulants, and can reflect alterations occurring in the CSF [34,123,124,125,126,127]. Among the most common AD biomarkers detected in saliva are Aβ peptides, total tau (t-tau), phosphorylated tau (p-tau), acetylcholine, and lactoferrin [128]. These salivary biomarkers may originate from nerves near the salivary glands, blood transferred to saliva, or oral bacteria causing chronic inflammation [34]. Additionally, compromised BBB, which is frequent in NDDs, can allow potential biomarkers to cross into the saliva [34]. This makes saliva a promising option for detecting AD biomarkers in a non-invasive manner.

#### 2.3.2. Findings of Salivary Biomarkers


A. Amyloid Pathology Biomarkers


The presence of Aβ peptides in saliva may result from the degradation of buccal cells, which share a common ectodermal origin with neuronal cells [126]. However, the mechanisms behind the formation and transport of Aβ to saliva are not yet fully understood [129].

Interactions between saliva, blood, and the degradation of buccal cells suggest that Aβ peptides should be detectable in saliva, especially since APP is widely expressed in peripheral tissues [128]. The close contact between the buccal mucosa and saliva further supports the idea that these changes could influence protein marker levels in saliva [126]. Recent studies have also identified elevated levels of salivary Aβ42 in AD patients, while Aβ40 levels remained unchanged, indicating that saliva could serve as a valuable biofluid for detecting specific AD-related biomarkers [128]. However, further research is required to validate the utility of salivary Aβ42 as a reliable marker for AD detection [128].

Several studies have shown an increase in Aβ42 concentration in the saliva of AD participants compared to healthy controls, suggesting that Aβ42 could serve as a potential screening marker for AD [10,34,125,127,129]. The difference in Aβ42 levels between CSF and saliva in AD patients may be attributed to Aβ42 production by multiple organs. This widespread production may result in increased levels of Aβ42 in saliva, even though CSF shows reduced levels of Aβ42 [130]. However, the use of salivary Aβ42 to distinguish AD patients from healthy individuals has demonstrated limited sensitivity (0.84) and specificity (0.68) [125], suggesting that further studies are needed to address existing gaps, uncertainties, and contradictions.

Acetylcholinesterase (AChE) is another biomarker that can be detected in saliva and has potential relevance for AD diagnosis [130,131]. As an enzyme primarily responsible for halting neurotransmitter activity in neuron signaling, AChE plays a significant role in AD pathology by contributing to the formation of Aβ fibrils and the development of amyloid plaques [128]. Current pharmacological treatments for AD often target AChE through inhibitors that prevent the breakdown of acetylcholine, thereby improving cognitive function [15]. Typically used for evaluating Aβ concentration in the blood, AChE can diffuse into saliva due to the innervation of the salivary glands by cholinergic neurons [130,131]. However, despite its potential, there is insufficient reliable data supporting the widespread use of salivary AChE as a biomarker for AD, and further research is needed to validate its clinical application [129].


B. Neuroinflammation Biomarkers


Lactoferrin (Lf) is an antimicrobial peptide that can bind to Aβ and has been proposed as a potential biomarker for brain infections linked to the development of AD [34,128,129]. Lf has been detected in the brain and CSF of AD patients, particularly within neurofibrillary tangles, amyloid plaques, and microglia [126,128,132]. Studies have shown that AD patients exhibit significantly lower levels of Lf compared to healthy controls, suggesting that Lf might be a promising biomarker for the early detection of AD [34,126,128,129].

Research demonstrates a positive correlation between Lf and Aβ42 and a negative correlation between Lf and t-tau, offering potential as a biomarker for identifying early-stage AD and MCI [34,128]. Decreased salivary Lf levels appear to be specific to AD and MCI, as this reduction is not observed in control groups, healthy elderly individuals, Parkinson’s disease patients, or other NDDs [132,133,134]. Salivary Lf composition can detect prodromal AD and AD dementia, distinguishing these conditions from frontotemporal dementia (FTD) with over 87% sensitivity and 91% specificity [132]. In the AD group, Lf levels also correlate with cognitive assessment scores on the Mini-Mental State Examination (MMSE) and changes in Aβ42 concentration in CSF, with 100% sensitivity and 100% specificity [129]. Overall, Lf has proven more accurate for diagnosis than CSF t-tau and Aβ42 and is considered a prognostic biomarker for the early diagnosis of AD [128,129].


C. Tau Pathology Biomarkers


Research suggests that p-tau can be detected in saliva and may serve as a biomarker for AD. The presence of p-tau in saliva is likely due to its proximity to the central nervous system, with p-tau potentially being released through nerves in the salivary glands, from acinar cells expressing high levels of tau mRNA, or from the breakdown of buccal cells [126]. The signaling pathways connected to protein kinases may be linked back to the inflammatory and neural-related AD symptoms [135]. Tau-tubulin kinase (TTBK) I and II are two examples of significant kinases [135]. The hyperphosphorylation of PHF-tau is caused by these enzymes, which phosphorylate tau protein at particular serine/threonine residues [135]. As a result, TTBKI changes native tau into PHF-tau, which is one of the main indicators of Alzheimer’s disease. P-tau phosphorylated at specific sites, such as threonine 181 (p-tau 181), serine 199 (p-tau 199), and serine 231 (p-tau 231), has shown greater specificity in distinguishing AD from other NDDs [129,131]. Notably, elevated p-tau 181 levels have been shown to correlate strongly with amyloid and tau PET imaging data, offering greater sensitivity and specificity compared to t-tau [129]. Studies demonstrated a significant increase in the p-tau 181/t-tau ratio in AD patients compared to control subjects, despite no differences in t-tau levels between the groups [129,136]. Overall, these studies indicate that p-tau, particularly p-tau 181, may serve as a more specific and reliable biomarker for AD compared to t-tau, offering improved accuracy in diagnosis and differentiation from other NDDs.


D. Oxidative Stress Biomarkers


The pathogenesis of AD is complex, with neuroinflammation and oxidative stress (OS) playing critical roles. Oxidative stress causes changes in the salivary redox balance, contributing to systemic imbalances observed in conditions such as AD [131]. OS occurs when there is excessive production of reactive oxygen species (ROS) and reactive nitrogen species (RNS), and the body is unable to neutralize them [131]. High levels of ROS and RNS cause oxidative damage to cellular components, resulting in dysfunction at both the cellular and organ levels and contributing to AD progression [131]. The increase in oxidative stress associated with AD is linked to reduced levels of the brain antioxidant glutathione (GSH) [137], with reduced GSH levels distinguishing patients with severe dementia from those with mild to moderate dementia [131].

Another key factor influencing redox imbalance in saliva is the stress hormone cortisol. Higher baseline cortisol levels in AD patients are associated with faster disease progression, suggesting that elevated cortisol exacerbates oxidative stress and inflammation [138]. In cognitively healthy older adults at risk for AD, elevated cortisol levels correlate with greater Aβ accumulation in the brain [138]. These findings suggest that stress-induced hormonal changes, alongside oxidative stress, may accelerate disease progression and contribute to the overall disruption of redox homeostasis in AD patients [138].

Glucocorticoid receptors in the hippocampus and prefrontal cortex mediate the effects of cortisol on cognitive function, including declarative memory and working memory [139]. Research has found that elevated cortisol levels in AD patients are often associated with a marked deficit in working memory relative to controls, a worse prognosis, and rapid cognitive deterioration [15,138,139,140,141].

Salivary cortisol concentrations are considerably greater in AD patients compared to healthy controls [139]. Studies have found a correlation between increased cortisol levels and decreased cognitive performance [140]. However, a separate meta-analysis, which included ten studies with a total of 2212 participants, found no significant difference in salivary cortisol levels between individuals with MCI and healthy controls [142]. Additional findings also indicated an absence of a relationship between salivary cortisol and the presence of AD. This inconsistency may be explained by the lack of specificity in measuring salivary cortisol, which may be due to antibody cross-reactivity with cortisone in saliva [143].

Sirtuins (SIRTs) have emerged as a critical focus in the study of brain aging and NDDs [123]. These proteins, which are members of the histone deacetylase family, play a vital role in regulating various biological processes through epigenetic mechanisms, such as gene expression and cell metabolism [15]. Elevated levels of SIRT1 in the hippocampus, a brain region crucial for memory, have been hypothesized to provide a protective mechanism against AD [144]. In contrast, decreased expression of SIRT6 in the neurons of AD patients suggests that this protein may also play a protective role in preventing AD pathogenesis [123]. Supporting this idea, a study showed a significant difference in SIRT6 expression between healthy individuals and those of advanced age, with the former showing a 2.5-fold greater expression area [123]. While SIRT5 levels remain relatively consistent between AD patients and healthy controls, there is a significant decrease in the expression of SIRT1, SIRT3, and SIRT6 in AD patients [15,123]. The decline in SIRT3 expression across multiple brain regions, along with disruptions in SIRT1 and SIRT6 signaling pathways, is believed to interfere with neural adaptability, exacerbating the progression of AD and the manifestation of its symptoms [123].


E. Epigenetic Biomarkers


MicroRNAs (miRNAs) are small, single-stranded genetic sequences, typically 21 to 23 nucleotides in length, that lack protein-coding information [126]. Despite their size, miRNAs play a crucial role in regulating gene expression by interacting with messenger RNA, ultimately influencing protein synthesis. These molecules are released into various bodily fluids, including blood, saliva, and urine, where their presence and expression patterns can be studied [126]. Abnormal miRNA expression profiles in the bloodstream can disrupt the production of specific proteins, leading to their accumulation in the brain. This makes miRNAs promising biomarkers for the detection of NDDs [145]. In AD, miRNAs have been implicated in the development of Aβ pathology by modulating the expression of APP and other key enzymes involved in Aβ processing, such as β-secretase [146]. Studies have reported a substantial increase in miR-455-3p levels within the serum of AD patients, a finding corroborated by post-mortem brain tissue analysis [145]. It was discovered that there was a significant reduction in miR-223, a miRNA linked to inflammation and potentially involved in central nervous system repair [147,148]. This decrease was closely associated with MMSE scores [145]. Table 3 provides an overview of the aforementioned saliva derived markers.

#### 2.3.3. Limitations

One of the limitations of using saliva as a biomarker for AD diagnosis is the low concentration of analyte, which requires highly sensitive and precise analytical methods. Additionally, the saliva collection procedure poses a challenge, as one-third of participants may be unable to provide an adequate sample, with research showing no statistically significant difference in Aβ42 protein concentrations between AD patients and non-AD individuals [34]. This suggests that various factors may influence the variability of Aβ42 protein levels in saliva, making it insufficient as a sole diagnostic marker.

Similar issues occur with cortisol, where inconsistencies in research results arise due to several factors, such as limited sample sizes, differences in testing methods, and gaps in scientific knowledge. Elevated cortisol levels, for instance, have been linked to personality traits, sleep patterns, mood states, and life stressors, though the precise mechanisms of these relationships remain poorly understood [140]. Overall, the inconsistent findings from salivary cortisol measurement highlight the need for further studies to refine diagnostic approaches [129].

## 3. Artificial Intelligence and Machine Learning

NDDs are often misdiagnosed due to overlapping symptoms, similar brain pathologies, and varied clinical presentations. This highlights the urgent need for reliable diagnostic biomarkers and methods to track disease progression, facilitate early diagnosis, and assess the effectiveness of new therapies and disease-modifying strategies. In response to the limitations of invasive procedures, recent research has shifted its focus toward non-invasive diagnostic and therapeutic approaches [150].

Although non-invasive methods hold great potential for early AD prediction, they also present computational challenges that must be addressed. Artificial intelligence (AI) and deep learning (DL) algorithms have the ability to process large amounts of data, identifying complex patterns that conventional analytical techniques may overlook. In AD, where subtle variations in data can signal the early stages of the disease, AI and DL techniques can play a crucial role in developing predictive models that support the early identification and management of the disease [151]. For instance, AI models trained on brain imaging data, such as MRI scans, can identify patterns of atrophy in brain regions indicative of AD, offering early indicators of the disease [151]. Additionally, these technologies may integrate multiple data sources, including biomarkers, genetic information, and lifestyle factors, to support early interventions and improve patient outcomes [151]. By analyzing a combination of blood biomarkers, genetic markers, and cognitive test results, AI systems can generate a comprehensive risk profile, enabling tailored treatment and monitoring strategies [152]. Additionally, AI and ML are revolutionizing Alzheimer’s diagnosis by using predictive modeling that surpasses conventional biomarkers. Predictive algorithms, for instance, may now use data from wearable technology to track changes in heart rate, sleep habits, and physical activity—all of which have been linked to cognitive decline [153]. AI provides real-time insights that can anticipate illness onset earlier than traditional approaches by combining these lifestyle factors with genetic and imaging data [153]. As a result, Alzheimer’s care is shifting toward more preventative and individualized interventions by enabling ongoing monitoring and empowering physicians to make proactive decisions based on a patient’s health data. Nevertheless, it is important to note that since AI depends on these conventional biomarkers and proven mechanisms to increase its ability to forecast as well as its performance in early identification of AD signs and symptoms, foundational investigations are still necessary to find novel biomarkers similar to miRNA that may enhance patient outcomes.

A deeper understanding of AD pathology can be achieved by integrating AI with advanced technologies such as Functional Near-Infrared Spectroscopy (fNIRS), Neurite Orientation Dispersion and Density Imaging (NODDI), and Magnetic Resonance Spectroscopy (MRS) [154,155,156]. Each of these modalities provides unique insights into the neurodegenerative processes associated with AD. For example, fNIRS captures hemodynamic response data, which are essential for understanding the functional deficits linked to AD [155]. NODDI excels in identifying microstructural alterations in white matter, providing information about neurite density and orientation that can be used to infer neurodegenerative processes [156]. MRS measures metabolic changes related to neuronal loss and dysfunction, such as reduction in N-acetyl aspartate (NAA), a marker of neuronal health [157].

These imaging modalities complement structural imaging modalities by uncovering novel biomarkers that may help improve the diagnosis of AD [154,155,157]. Combining data from multiple modalities may also result in more robust patient classification and diagnostic frameworks. Complex patterns and relationships that might not be visible through conventional analysis can be found by applying AI algorithms to these various datasets [82]. AI can potentially improve the identification of novel biomarkers by detecting minute changes in brain structure and function by integrating data from fNIRS, NODDI, and MRS [82,154]. These biomarkers can potentially increase diagnostic precision and offer a more complex understanding of AD pathogenesis. The volume of data available for analysis can significantly impact the accuracy of these predictive frameworks for diagnosing AD. The more data these models are exposed to, the better they become at identifying patterns and associations, enhancing their ability to detect and describe potential biomarkers with precision. Since blood-based testing, imaging techniques, and salivary extractions generate large and complex datasets that can be used to identify early AD biomarkers, the integration of these non-invasive methods with AI holds transformative potential for detecting and diagnosing early AD and improving patient outcomes [151].

## 4. Further Research

The complex relationships and interdependencies between Aβ42, t-tau, p-tau, NfL, and YKL-40 are crucial for enhancing the accuracy and precision of AD diagnosis [158]. A more comprehensive diagnostic picture is provided by the correlation between, for instance, higher t-tau and p-tau levels and lower Aβ42 levels. Additionally, increased NfL and YKL-40 levels, respectively, further support signs of neurodegeneration and inflammation, likely due to the underlying physiological pathways involved [45,52]. These interactions highlight the value of a comprehensive strategy that uses both established and new biomarkers to fully capture the range of disease pathology in AD diagnosis and comprehension. By addressing various facets of pathology, combining several biomarkers, such as Aβ42, t-tau, and p-tau, improves diagnosis accuracy in AD; for example, a greater indication of AD is provided by the combination of increased p-tau and decreased Aβ42 than by either biomarker alone, increasing early diagnosis sensitivity and specificity [45,52].

One promising approach is combining the Aβ42/Aβ40 ratio with other biomarkers using high-resolution liquid chromatography–tandem mass spectrometry (HR LC-MS/MS), which has demonstrated excellent discriminatory results, including an area under the curve (AUC) of 0.90 and an accuracy of 86%. Additionally, exploring the effectiveness of novel biomarkers across different stages of AD could provide a more comprehensive understanding of the disease. Longitudinal studies are essential in confirming the predictive power of these biomarkers, particularly in the preclinical phases of AD. Applying biomarker-based therapeutic strategies could help alter the disease’s trajectory. To achieve this, it is crucial to understand the molecular pathways through which these biomarkers influence AD pathogenesis, providing a foundation for future therapeutic interventions.

## 5. Conclusions

In this review, we provide a comprehensive overview of CSF biomarkers, blood-based biomarkers, and salivary biomarkers for diagnosing AD. While these biomarkers show significant potential, none have yet demonstrated sufficient precision to replace the costly and inefficient PET scans and CSF assays currently in use. Due to the absence of a single biomarker with the accuracy required to serve as a reliable alternative, further research is needed to identify novel or combined biomarkers that can improve the clinical efficacy of existing diagnostic tests.

New diagnostic techniques offer the promise of earlier and more affordable detection, which could lead to earlier intervention and enhanced treatment outcomes. CSF biomarkers have demonstrated promising potential for enhancing clinical practice by providing accurate diagnosis and valuable prognostic insights [157]. Blood biomarkers, in particular, represent a significant breakthrough, as they offer a minimally invasive and cost-effective method for diagnosing cognitive diseases [122]. By utilizing the association between AD biomarkers, a combined analysis is made possible that will undoubtedly identify a patient-specific profile for future AD diagnosis [158]. Nonetheless, further research remains necessary to confirm the utility of all AD biomarkers in clinical settings.

## Figures and Tables

**Figure 1 cells-13-01901-f001:**
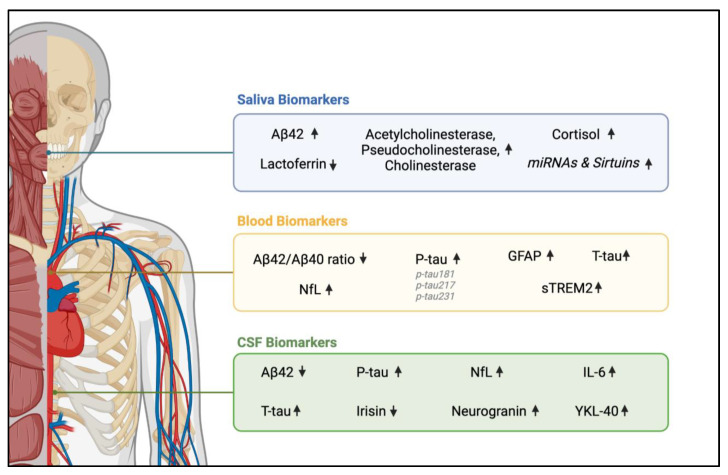
Overview of biomarkers in three biofluids associated with AD: saliva (blue box), blood (orange box), and Cerebrospinal Fluid (CSF) (green box). The arrows reflect an increase (upward arrow) or decrease (downward arrow) in each specific biomarker in the presence of AD, providing insight into the disease’s impact on various biological pathways across different fluids. Created in BioRender. Abiad, E. (2024) https://BioRender.com/z08y571 (accessed on 27 October 2024).

**Figure 2 cells-13-01901-f002:**
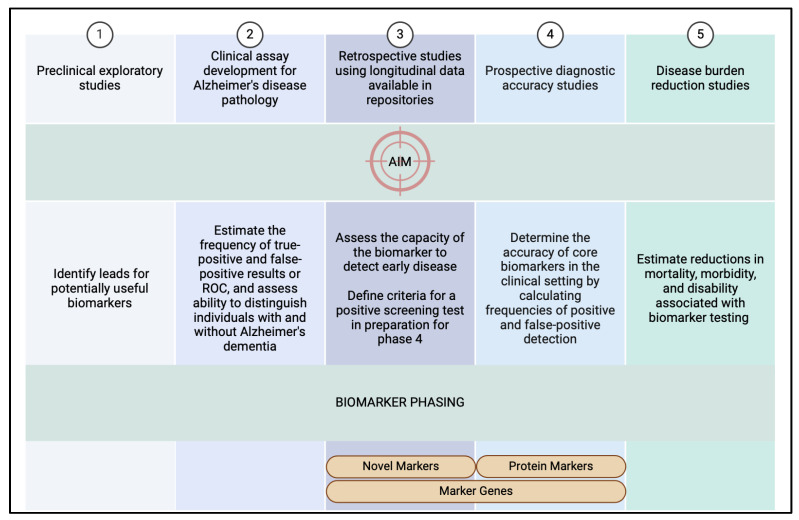
Overview of the 5-phase framework for AD biomarker development along with the primary aim of each phase. The figure highlights the stage of each biomarker category discussed in this section. Novel markers: GFAP, sTREM2, NfL. Protein markers: Aβ peptides and tau proteins. Marker genes: PSEN1 and PSEN2 and APOε4 [121]. Created in BioRender. Al-Aani, U. (2024), https://BioRender.com/x85d069 (accessed on 27 October 2024).

**Table 1 cells-13-01901-t001:** Overview of CSF Biomarkers in AD: The table summarizes key biomarkers, including their biological roles, diagnostic findings, and relevance across different stages of AD. It highlights the limitations and advantages of each biomarker, providing a comprehensive view of their utility in AD diagnosis and progression monitoring.

	Biomarker	Description	Finding	Target Stage	Limitations	Advantages	Reference
Traditional Protein Marker	Aβ	Aβ42 is the predominant form of the amyloid protein. Amyloid aggregation is a hallmark of AD.	Aβ42 levels are reduced in AD patients compared to non-AD individuals, and the Aβ42/Aβ40 ratio improves diagnostic accuracy.	All stages.	CSF patterns in different NDDs may be similar and not unique to AD, except for the Aβ40/42 ratio.	Aβ demonstrates 90% sensitivity and 80% specificity in differentiating AD from healthy individuals.	[20,21,24,26]
	T-tau	T-tau protein levels in CSF are hypothesized to indicate the degree of neuronal damage in AD.	T-tau levels are higher in AD patients compared to non-AD individuals and are directly linked to cell death and neuronal damage.	All stages.	Invasive lumbar puncture procedures carry risks of complications and standardization issues across laboratories.	High sensitivity (84%) and specificity (91%)	[27,30,34]
	P-tau	P-tau is elevated in AD, indicating tau pathology and neuronal degeneration.	P-tau levels are much higher in AD patients than in healthy individuals, demonstrating strong diagnostic power in differentiating AD.	All stages.	Similar to t-tau, CSF p-tau levels are not entirely unique to AD; correlation with other markers is necessary.	High sensitivity (90.2%) and specificity (80%)	[35,38,39]
Novel Markers	Irisin	Irisin is an exercise-induced hormone linked to AD and potentially correlates with Aβ42 and t-tau levels in CSF.	Reduced CSF irisin levels in AD patients correlate positively with Aβ42 and negatively with t-tau.	Clinical stage.	Limited research on sensitivity and specificity for AD; further investigation is needed.	Indicative of severity (promising prognostic marker)	[41,42,61]
	Neurogranin	Neurogranin is a postsynaptic protein elevated in the CSF of AD patients, indicating synaptic dysfunction.	It correlates with t-tau, p-tau, and the Aβ42/40 ratio and is associated with synaptic dysfunction in AD.	Late-stage AD.	Moderate sensitivity and specificity (0.73); it requires a combination with other biomarkers for accurate diagnosis.	It provides insight into synaptic dysfunction in AD and has the potential to enhance diagnostic precision.	[48,49,50]
Inflammatory Markers	IL-6	IL-6 is a pro-inflammatory cytokine involved in neuroinflammation and is elevated in the CSF of AD patients.	Elevated IL-6 levels correlate with AD, indicating neuroinflammatory mechanisms underlying AD.	MCI and early AD.	Limited specificity (76.9%) to AD; elevated IL-6 is observed in various NDDs.	High specificity (100%) in predicting AD; a potential marker for inflammation in AD.	[16,56]
	YKL-40	YKL-40 is a glycoprotein produced in astrocytes. It is elevated in CSF of AD patients and linked to neuroinflammation.	Elevated YKL-40 levels correlate with disease severity and progression in AD patients.	MCI and early AD.	Moderate sensitivity (65.6%) and specificity (66.3%); it requires a combination with other biomarkers for accurate diagnosis.	It reflects neuroinflammatory processes and has the potential to enhance AD diagnostic accuracy when combined with other markers.	[52,55]
Damage Markers	NfL	NfL is a biomarker of neurodegeneration and axonal damage; it is elevated in AD and other NDDs.	Increased NfL levels in CSF are associated with the severity and progression of AD; it can aid in distinguishing AD from other dementias.	Early clinical stages.	Lower sensitivity (59.6%) and specificity (76.2%) compared to traditional markers; Sensitivity and specificity were influenced by age and other conditions.	Indicator of neurodegeneration; correlates with clinical progression and disease severity.	[43,44,47]

**Table 2 cells-13-01901-t002:** Overview of Blood-Based Biomarkers in AD: The table summarizes key biomarkers, including their biological roles, diagnostic findings, and relevance across different stages of AD. It highlights the limitations and advantages of each biomarker, providing a comprehensive view of their utility in AD diagnosis and progression monitoring.

	Biomarker	Description	Finding	Target Stage	Limitations	Advantages	Reference
Marker Genes	APP	APP is a transmembrane protein highly expressed in neurons; it plays a role in synapse formation and neuronal plasticity.	Gene mutations promote the Aβ42/Aβ40 ratio; APP fragments in plasma are linked to neuronal death and the conversion of MCI to AD.	Early-onset AD.	Further investigation into the mechanisms is required.	Potential early detection through plasma analysis.	[68,69,70,71,72]
	PSEN1	PSEN1 encodes a component of the *γ*-secretase complex that processes APP.	Mutations alter Aβ42/Aβ40 ratios, cause tau hyperphosphorylation, and promote apoptosis, which are associated with cognitive decline and reduced hippocampal volume.	Early-onset AD.	Limited to specific mutations.	Strong relevance for familial AD.	[73,74,75,77]
	PSEN2	Encodes presenilin-2.	Gene mutations contributing to pathogenesis have been identified.	Early-onset AD.	Few mutations have been identified, with limited insight into mechanisms.	Potential in the detection of familial AD.	[78,79]
	APOε4	APOε4 is a glycoprotein involved in cholesterol transport, abundant in the central nervous system.	Increases the risk of AD onset, impairs Aβ clearance, and promotes tau fibril formation; linked to other AD markers.	Late-onset AD.	APOε4 detection does not necessarily correlate with AD.	Well established biomarker.	[80,81,84,85]
Protein Markers	Aβ	Aβ proteins, including Aβ40 and Aβ42, are produced by APP cleavage.	Reductions in the plasma Aβ42/Aβ40 ratio indicate amyloid pathology and show high accuracy in distinguishing AD from healthy controls.	All stages.	Levels are variable across assays.	Potential for early detection through minimally invasive methods.	[86,88,89,113]
	Tau	Tau is a microtubule-associated protein that plays a role in stabilizing neuronal microtubules.	Plasma p-tau levels, particularly p-tau217, increase in the early stages of AD, offering high diagnostic accuracy comparable to CSF biomarkers.	All stages.	Diagnosis may require the assistance of additional biomarkers.	Tau demonstrates 95% sensitivity and 90% specificity as well as specificity and strong correlation with AD progression.	[37,63,88,90,91,93,114]
Inflammatory Markers	GFAP	GFAP is a protein involved in cell structure and BBB; it is highly expressed in brain astrocytes.	Elevated in AD patients; correlates with Aβ and tau levels; predictive of cognitive decline in MCI and CU participants.	MCI and early AD.	Elevated levels are not specific to AD and may be modulated by neuroinflammatory conditions. In addition, GFAP demonstrates 79% sensitivity and 74.3% specificity for AD.	Early biomarker of astrocytic activation.	[98,99,100,101,102]
	sTREM2	sTREM2 is a soluble receptor involved in microglial clearance of Aβ and inflammatory signaling.	Elevated in MCI and AD; it correlates with cognitive decline and other AD biomarkers (i.e., CSF Aβ42).	MCI and early AD.	Not well established in clinical utility.	Early biomarker of microglial activation. sTREM2 demonstrates 81.8% sensitivity.	[63,103,104,105,106,107,115]
Damage Markers	NfL	NfL is a component of the neuronal cytoskeleton.	Increased levels indicate neurodegeneration, which correlates with AD pathology and poor cognition.	All stages.	Increased levels are not specific to AD. NfL demonstrates 79% sensitivity and 52% specificity.	Sensitive biomarker of neurodegeneration.	[44,94,95,96,97,116,117,118]

**Table 3 cells-13-01901-t003:** Summary of the major salivary biomarkers for AD as categorized by their biological mechanisms. The table describes the biomarkers’ roles in AD diagnosis, supported by significant findings relevant to their utility, their clinical target stage, and their advantages and limitations in comparison to other salivary biomarkers.

	Biomarker	Description	Finding	Target Stage	Limitations	Advantages	Reference
Amyloid Pathology	Aβ	Aβ proteins, including Aβ40 and Aβ42, can be identified in saliva samples post-degradation of buccal cells.	Salivary Aβ42 levels are significantly elevated in AD patients, showing approximately a twofold increase in AD patients compared to non-AD individuals.	Early preclinical stages of AD.	Salivary Aβ42 yields limited sensitivity (0.84) and specificity (0.68).	There is a positive correlation between the concentration of Aβ42 and the stage of AD. The Aβ42/Aβ40 ratio is a more precise indicator of the stage and severity of AD	[34,125,126,128,129]
	AChE	AChE is an enzyme that hydrolyzes the neurotransmitter acetylcholine into choline and acetate at neuromuscular junctions. It also contributes to the formation of Aβ fibrils.	Salivary AChE activity is considerably greater in AD patients compared to healthy individuals.	Mild and moderate stages of dementia.	Its effectiveness as a diagnostic biomarker for AD remains uncertain. There are insufficient reliable data to support the use of salivary AChE as a biomarker for AD.	Pharmacological therapy for AD involves targeting AChE.	[15,128,129,130,131]
Neuroinflammation	Lf	Lf is detected in the brain and CSF of AD patients, specifically within neurofibrillary tangles, amyloid plaques, and microglia.	Lf is considerably lower in AD patients. A study presented a positive correlation between Lf and Aβ42, while there was a negative correlation between Lf and t-tau.	Early preclinical stages of AD.	Negative correlation between Lf and t-tau.	It can detect prodromal AD and AD dementia, distinguishing these conditions from FTD with over 87% sensitivity and 91% specificity. A decrease in Lf is specific to AD.	[34,126,128,129,132,133]
Tau Pathology	Tau proteins (p-tau and t-tau)	Tau protein is associated with the formation of NFTs.	Tau protein is suggested as a potential marker for acute neuronal damage and progressive neurodegeneration.	Early preclinical stages of AD.	Elevated t-tau levels are found not to be specific to AD diagnosis. Differences in t-tau levels are not detected among AD patients, Parkinson’s disease patients, and controls.	P-tau can serve as a biomarker for AD. P-tau (181) levels correlate with amyloid and tau PET imaging data and are sensitive and specific compared to t-tau.	[126,129,136]
Oxidative Stress	Cortisol	Cortisol is the main glucocorticoid hormone produced by the adrenal cortex.	Salivary cortisol concentrations are considerably greater in AD patients compared to healthy controls.	Preclinical or prodromal stage of AD.	Lack of specificity in measuring salivary cortisol levels.	Elevated cortisol levels are often associated with a worse prognosis and rapid cognitive deterioration.	[15,138,139,140,141,142,143]
	Sirtuins	Sirtuins are members of the histone deacetylase family that use epigenetic mechanisms to regulate several biological processes, including gene expression and cell metabolism.	Have significant deviations from healthy controls. Some sirtuins are significantly elevated in AD, while others are significantly lowered.	MCI and early AD.	SIRT5 levels remain relatively consistent between AD patients and control subjects. Lack of information regarding sensitivity and specificity measuring salivary sirtuins levels.	SIRT6 is a relatively well-established sirtuin in AD diagnosis.	[15,123]
Epigenetic	miRNAs	miRNAs are single-stranded genetic sequences devoid of protein-coding information, typically spanning 21 to 23 nucleotides.	Numerous studies have been conducted on a wide range of miRNAs, but research has yet to yield conclusive interpretations regarding their use as serum biomarkers for AD.	Early onset of AD.	Further research is required.	There is a substantial increase in miR-455-3p levels within the serum of AD patients, which is closely associated with MMSE scores. miRNA demonstrated a sensitivity of 90.8% and specificity of 74.3% in diagnosing AD.	[126,145,149]

## Data Availability

No new data were created or analyzed in this study. Data sharing is not applicable to this article.

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
