# Peer review of "Navigating the Alzheimer’s Biomarker Landscape: A Comprehensive Analysis of Fluid-Based Diagnostics"

_cells, 2024, doi:10.3390/cells13221901_

Round 1
Reviewer 1 Report
Comments and Suggestions for Authors
PIs point out Alzheimer's disease (AD), that the lack of a single accurate biomarker underscores the need for further research to identify truly novel or existed combined biomarkers to enhance the clinical efficacy of existing diagnostic tests. In this context, artificial intelligence (AI) and deep-learning (DL) tools present promising avenues for improving biomarker analysis and interpretation, enabling more precise and timely diagnoses. It is believed that combining biomarker data with AI tools may offer a promising path toward revolutionizing the personalized characterization and early diagnosis of AD symptoms.
This review provides a comprehensive analysis of Fluid-Based diagnostics. I feel is the most compelling review. I do not have much opinions to say. Only a few points list below:
1. Pay more attention of Cut-off biomarkers, particularly for antibodies generated from so many specific phospho-Tau, such as 181, 217, 231……
2. As described above for phospho-Tau, such as 181, 217, 231……The phosphorylation state of tau is crucial for AD symptom development, as it directly correlates with cognitive decline. You may consider what kinds kinases get involved and how important for these kinases involved in AD symptoms.
3. How sensitivity, specificity and accuracy for AD biomarkers -in CSF and blood and their co-relation, even saliva?
4. The cost for PET image, and how accuracy for early stage
5. To make a diagram to achieve the Guideline of AD by international professionals
6. Basic research remains essential to discover novel biomarkers (such as miRNA) that can improve clinical outcomes, as AI relies on these validated markers and established mechanisms to enhance its predictive power and efficacy in early diagnosis of AD symptoms.
Reviewer 2 Report
Comments and Suggestions for Authors
This review provides an analysis of AD biomarkers, focusing on fluid-based diagnostics including cerebrospinal fluid (CSF), blood, and saliva biomarkers. It reviews their functions and roles in AD diagnosis, discusses their potential in early diagnosis and treatment, and highlights the promise of artificial intelligence (AI) and deep learning (DL) in enhancing biomarker analysis and interpretation. I think this review is valuable.
Some comments as follows:
1. In the tables provided by the authors, some have provided sensitivity and specificity data for the biomarkers, while others have not. It would be best to list them all out.
2. In the application of artificial intelligence and machine learning, it would be best to provide an example detailing its use in the AD diagnosis.
3. In the discussion section, please discuss how multi-biomarker analysis can improve the accuracy of AD diagnosis.
4. There are some issues with the format of the table, please make the necessary modifications. (such as table1, the first column on the left looks strange.)
